# Conditioned Media from Head and Neck Cancer Cell Lines and Serum Samples from Head and Neck Cancer Patients Drive Catabolic Pathways in Cultured Muscle Cells

**DOI:** 10.3390/cancers15061843

**Published:** 2023-03-19

**Authors:** Nicolas Saroul, Nicolas Tardif, Bruno Pereira, Alexis Dissard, Laura Montrieul, Phelipe Sanchez, Jérôme Salles, Jens Erik Petersen, Towe Jakobson, Laurent Gilain, Thierry Mom, Yves Boirie, Olav Rooyakers, Stéphane Walrand

**Affiliations:** 1Otolaryngology Head and Neck Surgery Department, CHU de Clermont-Ferrand France, 63000 Clermont-Ferrand, France; 2Human Nutrition Unit, INRAE, Auvergne Human Nutrition Research Center, Clermont Auvergne University, CHU de Clermont-Ferrand France, INRAE, UNH, 63000 Clermont-Ferrand, France; 3Biostatistics Department, CHU de Clermont-Ferrand France, 63000 Clermont-Ferrand, France; 4Anesthesiology and Intensive Care, Department of Clinical Science Intervention and Technology, CLINTEC, Karolinska Institutet, 141 86 Huddinge, Sweden; 5Division of Perioperative Medicine and Intensive Care, Karolinska University Hospital, 171 77 Huddinge, Sweden; 6Clinical Nutrition Department, CHU de Clermont-Ferrand France, 63000 Clermont-Ferrand, France

**Keywords:** head and neck cancer, conditioned media, muscle cells protein metabolism, patient serum

## Abstract

**Simple Summary:**

Cancer cachexia in head and neck cancer (HNC) is mainly due to a decrease in food intake, but other causal mechanisms could also be involved. The role of secreted factors from the tumor cells in driving cancer cachexia and especially muscle loss is unknown. In this way, we wanted to study both the action of secreted factors from HNC cell lines and circulating factors in HNC patients on skeletal muscle protein catabolism. We used a conditioned media model and mix of sera from cancer patients to analyze the in vitro metabolic response with primary myotubes. The same metabolic response was obtained with tumor-conditioned media and mix of sera from cancer patients. Patient plasma compounds produced specifically by the tumor seemed to have this effect. Our results indicated that the atrophy observed in HNC patients cannot be solely explained by a deficit in food intake.

**Abstract:**

Background: The role of secreted factors from the tumor cells in driving cancer cachexia and especially muscle loss is unknown. We wanted to study both the action of secreted factors from head and neck cancer (HNC) cell lines and circulating factors in HNC patients on skeletal muscle protein catabolism. Methods: Conditioned media (CM) made from head and neck cancer cell lines and mix of sera from head and neck cancer (HNC) patients were incubated for 48 h with human myotubes. The atrophy and the catabolic pathway were monitored in myotubes. The patients were classified regarding their skeletal muscle loss observed at the outset of management. Results: Tumor CM (TCM) was able to produce atrophy on myotubes as compared with control CM (CCM). However, a mix of sera from HNC patients was not able to produce atrophy in myotubes. Despite this discrepancy on atrophy, we observed a similar regulation of the catabolic pathways by the tumor-conditioned media and mix of sera from cancer patients. The catabolic response after incubation with the mix of sera seemed to depend on the muscle loss seen in patients. Conclusion: This study found evidence that the atrophy observed in HNC patients cannot be solely explained by a deficit in food intake.

## 1. Introduction

Head and neck cancers (HNC) account for half a million new cancer cases in the world each year [1]. The prevalence of HNC due to tobacco and alcohol use is decreasing, while the prevalence of HNC due to human papillomavirus (HPV) infection is constantly increasing [2,3,4]. Undernutrition is very common in patients with HNC [5,6]. The main predisposing factors for malnutrition in HNC patients are poor dietary habits (the patient population are typically tobacco and alcohol users), severity of tumor stage (high tumor volume) at the time of treatment due to late diagnosis, and swallowing disorders or even mechanical obstruction caused by the tumor itself. The decrease in food intake seemed to be the main issue for nutritional imbalance and, therefore, etiology of malnutrition and cachexia in HNC [7]. 

Cancer cachexia is characterized by a loss of lean body mass, particularly skeletal muscle mass. One in three cancer patients are thought to die due to cancer cachexia, although the rate varies with cancer type. Cancer cachexia increases the risk of surgical complications during cancer surgery, impairs patient quality of life, and decreases patient survival. Cachexia-driven loss of skeletal muscle mass is multifactorial, and cannot be reversed by conventional nutritional support only [8]. 

Skeletal muscle primarily has a contractile function but it also plays a key role in body metabolic homeostasis, as it is the main body reservoir of amino acids and a primary site of insulin-stimulated glucose transport and utilization. Muscle protein homeostasis is an interplay between two key processes: anabolism that promotes muscle protein synthesis and inhibits muscle protein degradation, and catabolism that inhibits muscle protein synthesis and promotes muscle protein breakdown. If the protein anabolism–protein catabolism balance is disrupted, it results in a net loss of muscle proteins, which can lead to cachexia. A number of catabolic or anabolic drivers were identified in both animal models and in humans. Low levels of anabolic hormones (testosterone, growth hormone, insulin-like growth factor (IGF-1)), altered insulin production and tissue sensitivity to insulin, high levels of myostatin and high levels of inflammatory cytokines (TNFα, IL-6) are considered the main endocrine-system hypo-anabolic factors likely to drive cancer cachexia [9]. Among the catabolic factors, significant activation of autophagy and ubiquitin proteasome pathways are thought to drive increased muscle catabolism in cancer cachexia [9]. 

Some cancers have a high prevalence of cachexia while others have a lower one [10]. This could be due to a decrease in food intake and/or nutrient absorption, especially in digestive tract cancer. However, these food-related mechanisms cannot entirely explain the high prevalence of cachexia in other cancers, such as lung cancer or HNC [11]. One under-researched hypothesis is that some tumor cells have the ability to produce and secrete compounds that have pro-catabolic and anti-anabolic effects on skeletal muscle. These secreted molecules could, thus, act from a distance of the tumor site and dysregulate protein metabolism, thus contributing to cachexia. One model used to test this hypothesis is to treat muscle cells with conditioned media (CM) made from tumor cells [12,13,14,15,16]. Data produced using this type of model argued that tumor cell-derived factors are able to induce muscle atrophy [17,18,19]. However, these results were challenged by a recent study showing that tumor secretions had no effect on muscle atrophy [20]. Furthermore, there is no data available for HNC tumors, despite the fact that almost 50% of HNC patients are in cachexia at the outset of management [5].

Here, we aimed to evaluate the effect of factors secreted from HNC cell lines or circulating in HNC patients on muscle metabolism. We also aimed to determine whether this change was the same with sera from HNC patients experiencing severe vs. mild muscle loss. For that purpose, we chose two well-known HNC cell lines (UT-SCC-60A, UT-SCC-5) [21,22,23,24,25] and carried out a clinical study to collect sera from cancer and non cancer patients. We hypothesized that some compounds in the secretome produced by HNC tumors or HNC cell lines would deregulate key drivers of protein metabolism within the skeletal muscle cell.

## 2. Materials and Methods

This paper includes results from in vitro experiments with cancer cell lines and from a clinical trial named MYOMEC. The MYOMEC trial was approved by the local institutional review board (‘Comité de Protection des Personnes Sud-Est’) and the French drug safety agency (ANSM—Agence Nationale de Sécurité des Médicaments) and was registered at ClinicalTrails.gov under number NCT03111771. All patients included in MYOMEC provided written consent to participate, and the study was performed in full compliance with the Declaration of Helsinki guidelines on medical research involving human participants.

### 2.1. Study Design and Population 

For the clinical study, patients aged 18 to 75 years old were enrolled at the Department of Otolaryngology–Head and Neck Surgery, Clermont-Ferrand University Hospital. Patients included were assigned into two groups: group 1 was formed by HNC patients and group 2 was a control group.

To be included in group 1, a diagnosis of HNC had to be clearly established. Only squamous cell carcinoma was included. Skin cancer and nasopharyngeal, nasal cavity, or salivary gland carcinoma were not included in this study. Group 2 was formed of noncancer patients operated at the otolaryngology unit for benign tumors (such as parotidectomy or thyroidectomy). Exclusion criteria for group 2 were presence of cachexia or a diagnosis of cancer on anatomopathological analysis post-surgery.

Exclusion criteria for both groups were: heart failure, respiratory failure (requiring long-term oxygen therapy), chronic renal failure (modification of diet in renal disease (MDRD) clearance < 60 mL/min), moderate or severe chronic obstructive pulmonary disease, and insulin-dependent diabetes mellitus. 

Participants included in the cancer group were recruited at panendoscopy, which is the cornerstone procedure for HNC assessment. Panendoscopy was performed at the beginning of management, before any nutritional support was engaged. By definition, at that time, the future treatment path chosen for the patient following tumor assessment was not yet known. The patients were classified on the basis of their muscle mass defined by CT scan. Two equal groups were formed: a first group with the lowest L3 muscle-mass index (L3MMI), i.e., severe sarcopenia (SS group), and a second group with no or only mild sarcopenia (MS group). The cut-off in our series was L3MMI = 46 cm^2^/m^2^, i.e., the median L3MMI in our cancer cohort.

General patient characteristics were collected: age, gender, medical records, tobacco and alcohol use (classified as current user, former user, or never user), usual treatment, tumor characteristics, i.e., site, divided into classes: oral, oropharyngeal, laryngeal, hypopharyngeal cancer, or carcinoma of unknown primary (CUP) of the neck, classified by the tumor-node-metastasis (TNM) staging (TNM 7th edition).

Two 5 mL blood samples were harvested just before the panendoscopy. Serum preparations were carried out immediately. The blood was allowed to coagulate for 60 min, and then, sera was obtained by centrifugation at 4000× *g* at 4 °C for 10 min. Blood samples were snap-frozen in liquid nitrogen and stored at −80 °C until analysis. 

### 2.2. Clinical and Biological Nutritional Assessment

The nutritional assessment included the measurements of usual weight (weight 3 months beforehand, in kg), current weight (kg), and height (cm). Skeletal muscle function was assessed by the short physical performance battery test (SPPB). The SPPB test comprises 3 tasks (balance, 4 m walk, and 5 chair stands) with a maximum score of 12. Impaired muscle function was defined as a score ≤8 [26,27]. Muscle strength was assessed via a handgrip strength test (mean of 3 measures on the dominant arm with the forearm bent at 90°, result in kg). Blood albumin (g/L), transthyretin (g/L), and C-reactive protein (CRP) were measured on the day of surgery. These measures were used to calculate the following scores:Nutritional risk index (NRI) = 1.519 × blood albumin + (current weight/usual weight) × 41.7;Percent weight loss over the previous 3 months = [(usual weight-current weight)/current weight] × 100;Body mass index (BMI) = current weight/height^2^ (kg/m^2^).

### 2.3. Skeletal Muscle Mass Assessment

Body composition was assessed by bioelectrical impedance analysis [28,29] including impedance, resistance, and reactance. Impedance values were measured at 5, 50, 100, and 200 kHz. Resistance and reactance values were measured at 50 kHz. These measures were used to calculate the lean mass index using Kyle’s equation and muscle mass index was calculated using Janssen’s equation. The cut-off values for low muscularity were ≤17 kg/m^2^ in men or ≤15 kg/m^2^ in women for the Kyle lean mass index [30], and ≤10.76 kg/m^2^ in men or ≤6.76 kg/m^2^ in women for the Janssen muscle mass index [31].

Skeletal muscle mass was also assessed by CT scan of the abdomen at the third lumbar vertebrae (L3 level) [32,33,34,35]. This CT scan was performed during the disease staging procedure prior to surgery. The method employed, which is described elsewhere [32], serves to determine L3MMI (cm^2^/m^2^). The malnutrition thresholds were set at 52.4 cm^2^/m^2^ for men and 38.5 cm^2^/m^2^ for women [32]. 

### 2.4. Serum Amino Acids Concentration

Amino acid concentrations were analyzed using a HPLC method described previously [36]. Briefly, serum samples were deproteinized in 3% 5-sulfosalisylic acid dihydrate (SSA) containing 200 µM norvaline as internal standard. Amino acids from the serum were analyzed using precolumn derivatization with ortho-phthaldialdehyde/3-mercaptoproprionic acid on an HPLC system (Waters 2690 Alliance system with Waters 474 fluorescence detectors; Waters^®^, Stockholm, Sweden). Serum amino acid concentrations were measured for glutamic acid, asparagine, serine, glutamine, histidine, glycine, threonine, 3-methylhistidine, citrulline, arginine, alanine, taurine, tyrosine, valine, methionine, tryptophan, phenylalanine, isoleucine, ornithine, leucine, lysine, and used to calculate serum essential amino acids (EAA), serum branched-chain amino acids (BCAA), serum total amino acids (TAA). BCAA was the sum of valine + leucine + isoleucine. EAA was the sum of tryptophan + threonine + phenylalaline + lysine + histidine + methionine + valine + leucine + isoleucine. TAA was the sum of all 21 amino acids measured. EAA-to-TAA ratio was calculated.

### 2.5. Plasma Elisa Test

Plasma levels of IL6, IL8, IGF1, GDF8, activin-A, follistatin, FGF-21, and GDF-15 were measured by enzyme-linked immunosorbent assay (ELISA) using a Raybiotech^®^ kit under conditions set by the supplier. Plasma levels of proteins were measured on the day of panendoscopy on patients who fasted for at least 6 h.

### 2.6. Cell Culture

Human primary myoblasts were obtained from Gibco^®^. The cell lines used were A11440 and A12555 (lot number 19F, 603, 597). The myoblasts were sub-cultured in T175 flask in growth media (Dulbecco’s Modified Eagle Medium (DMEM)/F12 GlutaMAX^TM^, Gibco^®^, Grand Island, NY, USA) supplemented with 10% fetal calf serum (FCS) certified heat-inactivated (Gibco^®^) and 1% antibiotic–antimycotic (ABAM, Gibco^®^) at 37 °C and 5% CO_2_. For experiments, myoblasts were detached at 70% confluence with TrypLE^TM^ (Gibco^®^) and transferred to 6-well plates coated with an attachment factor (Gibco^®^). Seeding density for the experiments was 85,000 cells per cm^2^. At 90% confluence, cell differentiation was induced in the myotubes using differentiation medium (DMEM/F12 Glutamax supplemented with 2% horse serum (Gibco^®^), 1% antibiotic/antimycotic (ABAM, Gibco^®^)). The medium was replaced every two days. Myotube differentiation was monitored using light microscopy. Myotubes used for experiments were at day 7 to day 10 of differentiation. Cells were washed once with phosphate-buffered saline (PBS) and were then incubated in serum-free media (DMEM F12 GlutaMAX^TM^) supplemented with 10% of patient serum (for patient serum experiments) or with 4% horse serum media supplemented with a 50% conditioned media from cancer cell lines (final concentration of 2% horse serum in the experimental media). Experiments with conditioned media were carried out in triplicate (n = 3) and experiments with patient serum were carried out in duplicate (n = 2).

Mouse C2C12 myoblast cells were purchased from the ATCC (American Type Culture Collection, Manassas, VA, USA). Myoblasts were cultured in a growth medium composed of DMEM containing 4.5 g/L glucose, 2.4 g/L sodium bicarbonate, 10% fetal bovine serum, 100 IU/mL penicillin, and 0.1 mg/mL streptomycin, and incubated at 37 °C in humidified air with 5% CO_2_. The medium was changed every other day to ensure growth until 90% confluence. Myotube formation was induced by changing the growth medium to a differentiation medium consisting of DMEM supplemented with 2% horse serum, 100 IU/mL penicillin, and 0.1 mg/mL streptomycin for 5 days before cell treatment. Passages between 4 and 10 were used for experiments. 

Tumor cell lines were gifted as a courtesy of Pr. Dalianis (Department of Oncology and Pathology, Karolinska Institutet, Stockholm, Sweden) with permission from Pr. Grenman (Otolaryngology, University of Turku, Finland). The cell lines used were UT-SCC-60A and UT-SCC-5. These cell lines were cultured in a T75 flask in growth media (DMEM/F12 GltuaMAX^TM^, Gibco^®^) supplemented with 10% FCS-certified heat-inactivated (Gibco^®^) and 1% ABAM (Gibco^®^) at 37 °C and 5% CO_2_. The cells were used for experiments at confluence. Differentiated human myotubes were incubated for 48 h with conditioned media or patient serum.

### 2.7. Myotube Morphology Analysis

Myotubes were photographed directly in the culture plates without fixation, using an AxioCam ERc5s digital camera coupled with an AxioVert.A1 microscope and ZEN 2.3 software (Zeiss, Germany). Myotube diameter was measured from three independent experiments on myotubes in each condition. For each myotube, three random measurements were performed along the length of the myotube (n = 3 measurements/myotube) using the ZEN 2.3 software, and the average of these three measurements was considered as a single value.

### 2.8. Generation of Tumor Cell-Conditioned Medium (TCM)

Tumor cell lines were cultured until confluence in T75 flasks with growth media. At confluence, the cells were washed three times with PBS and were then incubated with 7 mL of DMEM/F12 media supplemented with 1% ABAM without serum. Tumor-conditioned medium (TCM) was collected after 24 h, centrifuged at 3000 g for 5 min at room temperature to remove cell debris, sterilized by filtration (on a 22 µm filter), and stored at −80 °C until use. Control-conditioned medium (CCM) was made by incubating 7 mL of DMEM/F12 medium supplemented with 1% ABAM without cells and without serum for 24 h in a T75 flask, centrifuged at 3000× *g* for 5 min at room temperature, sterilized by filtration (on a 22 µm filter), and stored at −80 °C until use.

### 2.9. RNA Isolation, Reverse Transcription, and Quantitative Polymerase Chain Reaction (RT-qPCR)

Total RNA from the myotubes was isolated by the TRIzol method (Thermo Fischer Scientific^®^, Waltham, MA, USA) following the manufacturer’s protocol. RNA purity and quality were assessed by spectrophotometry (Nanophotometer, Implen^®^, München, Germany) and agarose gel electrophoresis. Complementary DNA (cDNA) synthesis was performed using iScript™ Reverse Transcription Supermix (BioRad^®^, Hercules, CA, USA). qPCR was performed using SsoAdvanced Universal SYBR Green Supermix (BioRad^®^). The gene expression assays were as follows: 3 µL cDNA, 1 µL specific primers, 6 µL nuclease-free water, and 10 µL SYBR Green Supermix. The gene expressions studied were: P62, LC3B, MurF1, TRAF6, Fox O, MafBx, DDIT3, Perilipin-3 (PLIN3), and IL6. To study the autophagy pathway, we measured the expression of LC3 and p62. The proteasome pathway was monitored by observing expression levels of the E3 ligases, MURF1, and atrogin1/MafBx. We also observed TRAF6 and FOXO3, two transcription factors involved in the regulation of autophagy and proteasome pathways. We measured the expression of DDIT, also known as CHOP, as a marker of endoplasmic reticulum stress. The IL-6 gene expression was used as a marker of inflammation and the expression of perilipin-3 (plin3) as a marker of intramuscular lipolysis. GAPDH expression was used as a reference control. Reactions were performed using a thermal cycler (CFX 96-Well Real-Time Cycler, BioRad^®^) using thermal cycles of 95 °C for 2 min and then, 40 cycles of 5 s at 95 °C, followed by 30 s at 60 °C. Cycle threshold (Ct) was normalized against GAPDH reference gene to give a relative normalized expression (Cq). These Cq values were compared between exposure to TCM and CCM. 

The primer pairs used for RT-qPCR were designed using Primer-Blast via PubMed and then, were purchased from Invitrogen (Waltham, MA, USA). These custom primer pairs are reported in Appendix B. Primers not listed in Appendix B were purchased directly from Invitrogen.

### 2.10. Protein Isolation and Western Blotting 

Myotubes were treated either with CCM or TCM from the two tumor cell lines (UT-SCC-60A and UT-SCC-5) for different durations. The cells at 7 days of differentiation were washed three times with PBS. The myotubes were then incubated in a 6-well plate with 1 mL of DMEM/F12 Glutamax supplemented with 4% horse serum (Gibco^®^) and 1% ABAM, added at 1 mL per well of TCM or CCM for 48 h. The cells were then scraped on ice with Laemmli 2X lysis buffer (Sigma-Aldrich^®^, St. Louis, MO, USA). Denatured proteins were then separated by SDS-page on a polyacrylamide precast gradient gel (mini-Protean TGX-Gel^®^, BioRad). After UV activation for further quantification, proteins were blotted on a polyvinylidene membrane (immobillon-FL; Millipore^®^, Burlington, MA, USA), and the membrane was UV-scanned for total protein quantification (BioRad^®^). Immunoblots were blocked 1 h with a blocking buffer (Odyssey blocking buffer^®^, LI-COR Bioscience, Lincoln, NE, USA) and then, were probed with the following primary antibodies: anti-myosin (1/1000, MF20, DSHB^®^), anti-MuRF1 (1/1000, no. NBP1-54939; Novus Biologicals^®^, Centennial, CO, USA), anti-atrogin-1 (1/1000; no. AP2041; ECM Biosciences^®^, Versailles, KY, USA), anti-LC3 (1/1000; no. 0231–100; Nanotools, München, Germany), anti-phospho-AKT (Ser 473) (1/1000; #9271; Cell Signalling^®^, Danvers, MA, USA), and anti-p62 (1/2000, no. H00008878-M01; Abnova^®^, Taipei, Taiwan). After several washes in PBS plus 0.1% Tween 20, the immunoblots were incubated with IRDye 800 CW or 680 LT (LI-COR Biosciences^®^). The membranes were then analyzed with an odyssey scan (LI-COR Bioscience^®^) or with ImageJ software 1.52 (NIH) for total protein quantification.

### 2.11. Quantification of Autophagy Flux

Myotubes were cultured in 6-well plates (Greiner Bio-One^®^) and incubated for 48 h at 37 °C and 5% CO_2_ with either CCM or TCM for the CM experiments and with a 10% mixed serum from cancer patients (with two different mixes: severe (SS group) or mild (MS group) sarcopenia (see below)) or control patients. At 6 h before the end of the 48 h incubation, half of the wells were incubated with 50 µM chloroquine (CQ), an autophagy inhibitor. Cells were then harvested as described in the protein isolation/immunoblotting method. Autophagy flux was calculated by the following calculation: autophagy flux = (LC3B2 expression with CQ − LC3B2 expression without CQ) × 100.

### 2.12. Proteasome Activity Measurements

After experimental myotube incubation, cells were lysed on ice with a homogenization buffer (50 mmol Tris-HCl/L, pH = 7.5, 1 mmol EDTA/L, 5 mmol MgCl_2_/L, 0.1 mmol dithiothreitol/L, 10% glycerol). After centrifugation and before measurement of protease activities, protein content was measured by spectrophotometry (Nanophotometer NP 80, Implen^®^, Germany). Chymotrypsin-like activity of the proteasome fraction was measured using the fluorogenic substrate SUC-LLVY-AMC [succinyl-Leu-Val-Tyr-7-amido-4-methylcoumarin; AMC)] (Sigma-Aldrich^®^) [37]. Then, 10 µL of the supernatant fluid (∼10 μg protein) was incubated in 100 μL of buffer (50 mmol Tris-HCl/L, pH 7.5, 1 mmol ATP/L, 5 mmol MgCl_2_/L, and 150 μmol LLVY/L) in microplates. Standard curves were prepared using the AMC. Fluorescence was measured continuously over 1 h at 37 °C on a SpectraMax^®^ i3X system (Molecular devices^®^) at λex = 380 nm and λem = 460 nm. Proteolytic activity was calculated from the increment of the curves from samples and standards and were expressed as pmol of AMC released/μg protein per minute.

### 2.13. Statistical Analysis

All statistical analyses were performed using Stata software (version 13, StataCorp^®^, College Station, TX, USA). Significance was set at a two-sided type I error of 5%. Patient characteristics were presented as mean ± standard deviation (SD) or median [interquartile range] for continuous data (assumption of normality assessed using the Shapiro–Wilk test) and as number of patients and associated percentages for categorical variables. Quantitative variables were compared between independent groups (control vs. cancer) by a Student *t*-test or a Mann–Whitney test if the conditions of the *t*-test did not hold ((i) normality and (ii) homoscedasticity studied via a Fisher–Snedecor test). Between-group comparisons were performed using a chi-square test or, when appropriate, Fischer’s exact test for categorical variables. Relationships between quantitative outcomes were studied using correlation coefficients (Pearson or Spearman according to statistical distribution). Šidák correction was applied to address the problem of multiple comparisons. Finally, in paired contexts, we applied the usual appropriate statistical tests: paired Student test or a Wilcoxon test according to the assumptions of the *t*-test.

## 3. Results

### 3.1. Epidemiological Characteristics 

The MYOMEC study included 34 patients. The key epidemiological characteristics of the patients are summarized in Table 1 and Table 2. 

The cancer group was mainly composed of male patients, aged mostly between 50 and 70 years old, with an average BMI of 22.0 ± 3.2 kg/m^2^. Cancer-group patients had lost on average 6% of their usual weight in the last 3 months, and were mostly in a state of malnutrition, with an average L3MMI of 43.5 ± 7.8 cm^2^/m^2^ and an average NRI of 93.9 ± 10.9. All the sub-localizations of HNC were represented in our population (Table 2). 

The control group was mostly composed of younger female participants, with a stable weight and a tendency to be overweight (BMI = 30.4 ± 7.0). The difference in the recruitment, especially concerning gender, was due to our department’s surgical activity, as the control group was mainly operated for a benign tumor of the thyroid (Table 3).

### 3.2. Tumor CM Drove Atrophy on Differentiated Myotubes

The first step was to test the effect of the secretions of HNC cell lines on C2C12 myotubes after 48 h of incubation with conditioned media. The tumor-conditioned media (TCM) clearly induced C2C12 atrophy compared to control-conditioned media (CCM) treatment. The average myotube diameter was 23.1 ± 3.4 µm after incubation with TCM (n = 324) vs. 27.0 ± 3.8 µm after incubation with CCM (n = 231; *p* < 0.0001; Figure 1A–C). This result was mainly due to the presence of a higher number of myotubes with a small average diameter after incubation with TCM (Figure 1D). We investigated whether the change in myotube average diameter (assessed by microscopy) corresponded to a change in myotube myosin content (assessed by Western blot analysis). The myosin expression was 46% lower in the myotubes incubated with TCM compared to CCM (137 vs. 74 A.U.; CCM vs. TCM; *p* = 0.01; Figure 1E).

In response to these results and our primary aim, we explored the effect of TCM in human-differentiated myotubes. We found a 55% decrease in myosin content in human-differentiated myotubes after 48 h incubation with TCM (5.9 ± 1.4 vs. 2.7 ± 0.7 A.U; CCM vs. TCM; *p* <0.001; Figure 2). Given the more physiological aspect of the human myotubes, we decided to continue the experiments with human myotubes.

#### 3.2.1. Tumor CM Disrupted the Catabolic Pathway and Its Regulation

Compared to controls, human myotubes treated with TCM had lower p62 gene expression (0.92 ± 0.06 vs. 0.68 ± 0.20 A.U.; CCM vs. TCM; *p* = 0.04) but no difference in LC3 gene expression (0.98 ± 0.01 vs. 1.04 ± 0.25 A.U. for CCM vs. TCM; *p* = 0.73; Figure 3A). We found an overall decrease in proteasome gene expression in muscle cells cultured with TCM (Murf1 = 1.1 ± 0.07 for CCM vs. 0.3 ± 0.2 for TCM; *p* < 0.001; Fox O = 1.1 ± 0.07 vs. 0.6 ± 0.3, *p* = 0.002; TRAF6 = 1.0 ± 0.1 vs. 0.7 ± 0.2, *p* = 0.04; MafBx = 0.8 ± 0.2 vs. 0.9 ± 0.2, *p* = 0.4) (Figure 3B). However, compared to incubation with CCM, incubation with TCM led to no change in DDIT3 expression (0.88 ± 0.16 vs. 1.20± 0.42, *p*= 0.25), a decrease in PLIN3 expression (1.22 ± 0.31 vs. 0.82 ± 0.15, *p* < 0.001), and an increase in IL6 expression (0.02 ± 0.003 vs. 0.14 ± 0.06, *p* < 0.001) (Figure 3C).

As already shown [38], the decreased proteasome gene expression seen in muscle cells exposed to TCM could be due to feedback control of gene expression after high activation of the proteasome during the first hours of incubation. To test this idea, we ran an enzymatic time–course analysis of proteasome activity. As expected, we found that TCM was able to activate the proteasome after a short period of exposure, i.e., 6 h, although this change did not reach statistical significance (+37.5%, *p* = 0.16). Interestingly, after a sharp increase in proteasome markers at 6 h of incubation, the level of proteasome gene expression fell rapidly up to 20 h of incubation with TCM (Figure 3D).

#### 3.2.2. TCM Induced the Autophagy/Lysosome Pathway 

There was significant activation of the autophagy/lysosome pathway after 48 h incubation with the TCM compared with the CCM (Figure 4).

Taken together, the early proteasome activation followed by a later increase in autophagic activity likely explain the decrease in myotube diameter. 

### 3.3. Sera from Cancer Patients Were Not Able to Drive Atrophy in Differentiated Myotubes but Changed Myotube Metabolism in the Same Way as Conditioned Media

As the hypothesis was that the muscle loss seen in cancer patients is partially due to tumor secretions, we ran the same experiments but using patient sera from the MYOMEC study.

We did not observe any significant difference in myosin content after incubation with either a mix of sera from cancer patients or a mix of sera from control patients. The mean myosin level was 16.70 ± 0.20 A.U. after incubation with the control-group mix and 21.42 ± 6.00 after incubation with the cancer-group mix (*p* = 0.23, Figure 5). We investigated whether a mix of sera from patients with the highest muscle loss was able to induce atrophy, but we did not find any difference between the mix of sera from cancer patients with the highest muscle loss (SS group) vs. patients with less muscle loss (MS group) (Figure 5).

#### 3.3.1. Autophagy Genes

The incubation of differentiated myotubes with sera from cancer patients led to a decrease in both LC3 expression and p62 expression (LC3 = 1.54 ± 0.76 for the control group vs. 0.94 ± 0.14 for the cancer group, *p* = 0.01; p62 = 0.90 ± 0.26 for the control group vs. 0.54 ± 0.23 for the cancer group, *p* = 0.008). The decrease in gene expression seemed to be dependent on the magnitude of muscle loss, with lower gene expression in the case of severe muscle loss (p62 = 0.93 ± 0.45 for the MS group vs. 0.6 ± 0.34 for the SS group, *p* = 0.005; LC3 = 0.98 ± 0.18 for the MS group vs. 0.91 ± 0.10 for the SS group, *p* = 0.43) (Figure 6A). Taken as a whole, the results on autophagy gene expression in differentiated myotubes obtained with patient sera were similar to the results obtained with TCM.

#### 3.3.2. Proteasome Genes

As already observed with TCM, incubation of differentiated myotubes with sera from cancer patients decreased proteasome gene expression in the differentiated myotubes. All gene expressions were repressed or tended to be lower (Figure 6B): Murf1 = 0.87 ± 0.64 for the control group vs. 0.59 ± 0.43 for the cancer group, *p* = 0.14; MafBx = 1.41 ± 0.47 for the control group vs. 0.75 ± 0.11 for the cancer group, *p* = 0.002; TRAF6 = 1.01 ± 0.29 for the control group vs. 0.98 ± 0.24 for the cancer group, *p* = 0.85; Fox O = 1.25 ± 0.31 for the control group vs. 0.98 ± 0.30 for the cancer group, *p* = 0.08. In addition, the decrease in proteasome gene expression was linked to the magnitude of muscle loss in cancer patients. When we reviewed the effect of muscle mass, we observed that proteasomal gene expression was reduced in myotubes incubated with sera from severely sarcopenic HNC patients, as found for TCM (Murf 1 = 0.31 ± 0.26 for SS vs. 0.80 ± 0.43 for MS, *p* < 0.001; MafBx = 0.81 ± 0.14 for SS vs. 0.68 ± 0.03 for MS, *p* = 0.02; TRAF6 = 0.85 ± 0.19 for SS vs. 1.12 ± 0.24 for MS, *p* = 0.01; Fox O = 0.79 ± 0.17 for SS vs. 1.18 ± 0.25 for MS, *p* < 0.001 (Figure 6)).

#### 3.3.3. Endoplasmic Reticulum Stress/Lipid Metabolism/Inflammation Genes

PLIN3 gene expression increased after incubation with a mix of sera from cancer patients (PLIN3 = 0.72 ± 0.08 vs. 0.89 ± 0.08 (*p* < 0.001) for the control group and cancer group, respectively). There was a decrease in DDIT3 expression (DDIT3 = 1.56 ± 1.0 vs. 0.89 ± 0.25 for the control group and the cancer group, respectively, *p* = 0.01) and no change in IL-6 expression between the control group and cancer group sera mixtures (IL6= 0.51 ± 0.19 vs. 0.55 ± 0.20, *p* = 0.7). However, we found results that diverged from our previous findings with TCM, concerning the effect of sera selected according to the extent of patient muscle mass loss. With patient sera, IL-6 expression increased according to muscle loss (IL-6 = 1.07 ± 0.16 for the SS group vs. 0.47 ± 0.15 for the MS group, *p* < 0.001). PLIN3 gene expression increased with increased muscle loss (PLIN3 = 0.93 ± 0.09 for SS vs. 0.85 ± 0.04 for MS, *p* = 0.001) whereas DDIT3 showed no change (DDIT3 = 0.84 ± 0.17 for SS vs. 0.94 ± 0.40 for MS, *p* = 0.35) (Figure 6C). 

#### 3.3.4. Proteasome Activity

There was no change in proteasome activity after muscle cell treatment with sera from control vs. cancer patients (11.64 ± 2.01 pmol/min/µg for the control group vs. 12.69 ± 3.26 pmol/min/µg for the cancer group, *p* = 0.7; Figure 7A). However, proteasome activity significantly increased in patients with low muscle loss (14.97 ± 1.06 pmol/min/µg) but decreased in patients with severe muscle loss (10.41 ± 3.1 pmol/min/µg) (*p* < 0.001; Figure 7A). 

#### 3.3.5. Autophagy/Lysosomal Pathway

The autophagy/lysosomal pathway was activated in differentiated human myotubes after incubation with serum from the SS group (mean value 0.9 ± 0.4), but there was no difference between MS sera and control sera (0.5 ± 0.2 for MS vs. 0.6 ± 0.1 for control; Figure 7B).

#### 3.3.6. Synthesis of the Results

Finally, we produced a synthesis table of the effect seen with the tumor-conditioned media and the serum-conditioned media on human-differentiated myotubes. The results are shown in Table 4.

### 3.4. Blood Parameters That Could Explain the Serum Effect

A recent study established a link between high serum essential amino acids and skeletal muscle depletion in gastrointestinal cancer cachexia [39]. Moreover, much evidence suggests that inflammatory cytokines are key factors of muscle catabolism [40]. We, therefore, investigated the levels of these factors in our patient sera. We found no statistically significant change in blood levels of C-reactive protein, total protein, total cholesterol, triglycerides, HDL, TSH, GDF-15, FGF-21, Testosterone, IL-6, IL-8, Follistatin, and IGF-1, including between SS-group and MS-group sera. The results are reported in Appendix A. We found no correlations between amino acid concentrations and muscle mass.

## 4. Discussion

Muscle loss is very frequent at the outset of HNC management, with almost 60% of patients having muscle wasting, i.e., cachexia [5]. Cachexia in HNC patients is mainly due to malnutrition, as HNC causes pain, odynophagia, anorexia, and other nutritional impact symptoms (NIS) that decrease food intake [7]. The main NIS is reported to be dysphagia, as HNC patients with dysphagia experienced 5% body weight loss 45 days earlier than those without difficulty swallowing [7], but anorexia, pain, and mouth sores are also important explanatory factors. Other factors, such as tumor secretions, could also contribute to weight loss and especially muscle loss in HNC patients. However, to the best of our knowledge, there is no data on the potential effect of HNC tumor secretions on regulation of protein metabolism in skeletal muscle. Therefore, the main objective of the present work was to assess in vitro whether tumor secretions from HNC can contribute to muscle atrophy. 

We chose to include only patients at the outset of management in order to decrease the potential bias of treatment (surgery/radiotherapy/chemotherapy) effects on nutritional status. The patients were classified into subgroups according to degree of muscle loss based on L3MMI, i.e., a group with severe muscle loss (SS group) and a group with no or mild muscle loss (MS group). We found a higher proportion of cachectic patients with tumor stage T3-4 compared to non-cachectic patients (73% vs. 55%, not significant). This observation supports the fact that the pathogenesis of cancer cachexia in HNC could be a nutritional issue, because the presence of a larger tumor in T3-4 patients could contribute to dysphagia and upper digestive tract obstruction, leading to decreased food intake and weight loss. However, in the population studied in this study, food intake, dysphagia, and tumor stages were not different between the SS group and the MS group (Saroul N. CHU Clermont-Ferrand, INRAE, UNH, Université Clermont Auvergne 63000 Cler-mont-Ferrand, France. PhD thesis. Université Clermont Auvergne. 2021). Taken together, these data argue that skeletal muscle loss in HNC cannot be explained only by a decrease in food intake and subsequent low nutrient absorption. Another hypothesis is that a larger tumor is able to secrete more active factors involved in weight loss, contributing to greater muscle loss in patients suffering from a T3-4-stage HNC.

Many studies focused on potential biomarkers of cancer cachexia [41,42,43,44,45,46,47,48,49,50]; however, only one of these studies included HNC patients [45]. The main blood biomarkers of cachexia found in these studies were pro-inflammatory proteins (CRP, IL-6, IL-8, IL-10, IFNγ, TNFα), members of the TGFβ family (myostatin, activin), tumor-derived factors (ZAG, midkine), markers of lipolysis (leptin, adiponectin), and other proteins such as IGF-1, albumin, and angiotensin II. None of these studies defined the cachexia state on the basis of skeletal muscle loss but considered only weight loss. Here, none of the biological markers measured were clearly associated with either weight loss or muscle loss (Saroul N. CHU Clermont-Ferrand, INRAE, UNH, Université Clermont Auvergne, 63000 Cler-mont-Ferrand, France. PhD thesis. Université Clermont Auvergne. 2021.). We only found trends towards increased blood CRP, white blood cells, GDF-15, IL-8, and FGF-21, and decreased in testosteronemia in SS patients compared to MS patients. This could be due to the small sample size of our population. Note, however, that blood IL6 concentrations in SS (37 pg/mL) and MS patients (42 pg/mL) were higher than those measured in similar studies regarding HNC, with the mean blood level in the study population of IL6 varying from 1.35 to 9.7 pg/mL [51,52,53,54]. Il-6 is known to be increased in cancer cachexia and to correlate with weight loss in pancreatic cancer patients [55], and anti-IL-6 drugs are able to diminish muscle loss in preclinical models [56]. It was previously shown that HNC tumors are able to secrete IL-6 [57]. The very low level of IGF-1 in our study compared to normal values was another interesting result [58]. It was previously demonstrated that blood IGF-1 was reduced in oral cancer, cancer cachexia, and fasting [59,60,61]. Low IGF-1 is probably due to degraded nutritional status. IGF-1 is a key driver of protein metabolism. This hormone was shown to stimulate protein synthesis and inhibit protein degradation, especially in skeletal muscle, and decreased circulating IGF-1levels were associated with cancer cachexia [62]. Therefore, a reduction in plasma IGF-1 concentrations may have contributed to the effects of conditioned media from HNC plasma on muscle cells.

Because the effect of tumor secretion on muscle atrophy is still under debate, we set out to evaluate the effect of tumor secretion on a simple easily-cultivable model of mouse-differentiated myotubes (C2C12 cell line). C2C12 cell lines is a commonly used model to study muscle cell metabolism in vivo, despite the fact that this is a non-human cell line [63]. With this model, we observed that tumor secretions from HNC cell lines are able to induce atrophy in skeletal muscle cells. This observation was then confirmed in a more physiological model of human primary myotubes, where we observed a 50% decrease in myosin content after incubation with TCM. 

The next step was to find out the muscle cell pathway involved in the ability of cancer cell line secretions to induce atrophy. Our results support the idea that tumor secretions likely activate the proteasome and autophagy/lysosome pathways in a time-dependent manner, probably with early activation of the proteasome pathway and longer-term activation of the autophagy/lysosome pathway. This long-term activation of the autophagy process in our TCM model was already reported in vitro, where it was linked to IL-6 trans-signaling [64]. In this in vitro model [64], autophagy flux continued to increase after 3 days of myotubes incubation with a TCM. We obtained the same result after incubation of differentiated myotubes with both cancer-cell-conditioned media and cancer-patient sera. Another important result was the repression of key genes of the ubiquitin/proteasome system in both cancer-cell-conditioned media and cancer-patient sera. To the best of our knowledge, there are no other data available on the decrease in ubiquitin/proteasome gene expression after long-term incubation with TCM. However, in most publications, ubiquitin/proteasome-system gene expression increased during the first hours of incubation and decreased thereafter [38,65,66,67,68]. For instance, Zhang et al. reported that, at 24 h, mRNA levels of MafBx were lower in cells treated with cancer-patient CM than cells treated with control-subject CM [38]. Taken together, these data suggest that ubiquitin/proteasome gene expression is activated in a time-dependent manner in muscle cells treated with media containing tumor secretions. One of the mechanisms that may explain the reduction in proteasome activity could be the energy demand induced by the proteolytic system. The proteasome system demands much energy in the form of ATP. We observed a reduction in expression levels of key mitochondrial genes in muscle cells treated with serum from HNC patients (Saroul N. CHU Clermont-Ferrand, INRAE, UNH, Université Clermont Auvergne 63000 Cler-mont-Ferrand, France. PhD thesis. Université Clermont Auvergne. 2021.). One of the compounds present in the plasma of cancer patients could impair the production or consumption of ATP by muscle cells, thus limiting the activity of the proteolytic system [14]. Note that both the mix of sera from severely sarcopenic HNC patients and the CM from tumor cell lines were able to induce this decrease in proteasome-system gene expression. Our results also showed that the genes from the ubiquitin/proteasome system were not always involved in the atrophic program in the same way. This differential activation was described previously, with some studies finding a significant increase in atrogin1/MAFbx gene expression without change in Murf1 gene expression, whereas other studies found an activation of Murf 1 without changes in atrogin1/MAFbx gene expression [69,70,71].

Another explanation for muscle atrophy in HNC patients is a decrease in muscle protein synthesis, as seen in CM-treated myotubes. Decreased muscle protein synthesis is a hallmark of cancer cachexia and was observed in both human and mice models [72,73]. Measuring the phosphorylation state of the different proteins of the Akt/mTOR pathway can be challenging in terms of the timing of sampling after the anabolic stimuli and necessitates more complex experimental preparation with the need for an insulin and amino acid restriction period before stimulation. This is why the actual measure of protein synthesis with the use of a stable isotope or via the sunset method is the best method to monitor protein synthesis. However, in our study, we did not measure the protein synthesis rate and only measured the phosphorylation state of AKT at the serine 473, as AKT/mTOR is the main anabolic pathway regulating muscle protein synthesis [40,74]. We found that the TCM was able to decrease the phosphorylation of AKT, but we did not observe any difference between the control and HNC patient groups (both SS and MS; Saroul N. CHU Clermont-Ferrand, INRAE, UNH, Université Clermont Auvergne 63000 Cler-mont-Ferrand, France. PhD thesis. Université Clermont Auvergne. 2021). Therefore, some compounds secreted by tumor cells or the low IGF1 level measured could have downregulated the Akt/mTOR pathway, leading to a decrease in protein synthesis. However, more data are needed to understand the role of tumor-secreted factors on muscle protein synthesis. 

Our study had some limitations that must be addressed. In our conditioned media experiments, we used a no-cell-control approach, a better control would have been to expose the myotubes to conditioned media made after incubation for 24 h hour, with a non-tumorigenic cell line corresponding to squamous cell carcinoma. With our approach, it was difficult to confirm that the effect of the TCM was only due to tumor secretions and not by the depletion of the media component by tumor cells during the 24 h incubation. However, the TCM was produced from the tumor CM diluted with fresh media and supplemented with 2% serum, as we used when we differentiated the myotubes with a media change every 48 h. The second limitation of this study was that our control group did not match with the patient group in terms of age, sex, and body composition. Our results comparing control and cancer group could, therefore, have been an effect of the difference in hormone and metabolic profile between the two groups. Due to this reason, we chose to investigate, within our cancer group, the impact of patients’ muscle mass status on the in vitro effect of their serum on human myotubes. Furthermore, we observed that our most intriguing results were between the mild sarcopenic and severely sarcopenic HNC patients, where no difference in age, sex, and body composition could explain the differences observed. The third limitation of our study was the comparison between experiments using tumor-conditioned media and patients’ sera. For the patients, we used a concentration of 10% serum, a concentration much higher than the concentration of fetal bovine serum (2%) used during the tumor CM experiments. This might have been a confounding factor, which would explain the discrepancy observed between the impact of TCM on myotubes atrophy that was not observed with patients’ sera.

## 5. Conclusions

This study showed, for the first time, that cancer cell lines from HNC were able to induce atrophy in differentiated myotubes. Our results indicated that the atrophy observed in HNC patients cannot be solely explained by a deficit in food intake. It is now necessary to dissect the secretome of tumors in HNC patients, in order to find new biomarkers of deregulated muscle metabolism in cancer patients and to define new therapeutic approaches and targets that can help reduce cancer cachexia in HNC patients.

## Figures and Tables

**Figure 1 cancers-15-01843-f001:**
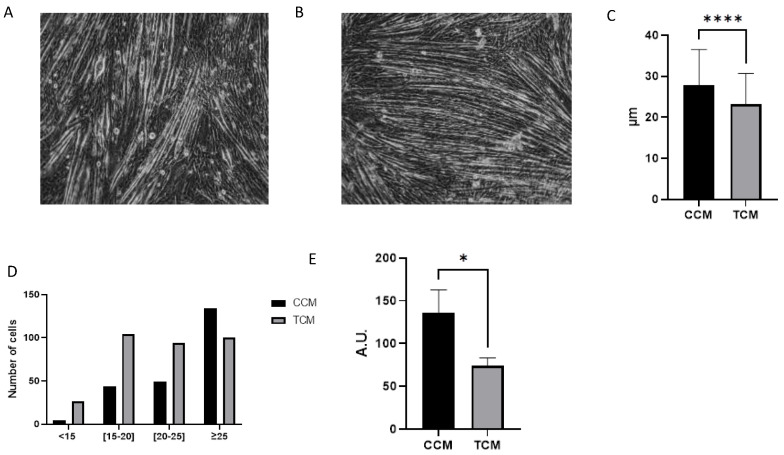
Effect of tumor cell-conditioned media or control-conditioned media on size and myosin content of C2C12-differentiated myotubes. (**A**) Picture of differentiated myotubes after incubation for 48 h with control-conditioned media (CCM). (**B**) Picture of differentiated myotubes after incubation for 48 h with tumor-conditioned media (TCM). (**C**) Average diameter analysis by microscopy of myotubes incubated with either CCM or TCM for 48 h (n = 231 for CCM and n = 324 for TCM), ****: *p* < 0.0001. (**D**) Number of myotubes classified according to their average diameter assessed by microscopy (in µm) stratified by media (CCM or TCM); <15: average myotube diameter <15 µm, [10,11,12,13,14,15,16,17,18,19,20]: average myotube diameter between 15 µm and 20 µm, [20,21,22,23,24,25]: average myotube diameter between 20 µm and 25 µm, ≥25: average myotube diameter ≥ 25µm. (**E**) Mean ± SD of myosin expression (heavy chain) assessed by Western blot analysis in C2C12 myotubes incubated with either TCM (n = 3) or CCM (n = 3).*: *p* < 0.05. AU: absorbance unit. The uncropped blots and molecular weight markers are shown in Appendix A.

**Figure 2 cancers-15-01843-f002:**
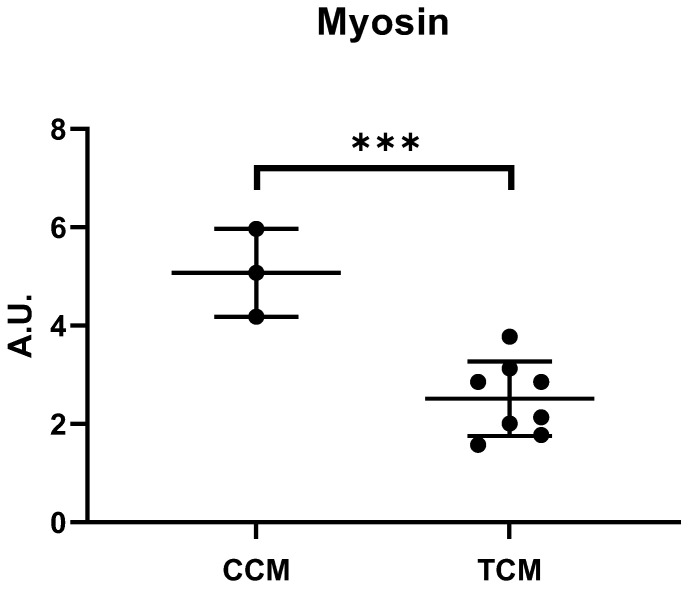
Myosin expression (heavy chain) in human primary myotubes incubated with either tumor conditioned media or control conditioned media. Mean ± SD of (heavy-chain) myosin expression levels in myotubes incubated with either tumor-conditioned media (TCM) (n = 3) or control-conditioned media (CCM) (n = 3, two tumor cell lines). ***: *p* < 0.001. AU: absorbance unit. The uncropped blots and molecular weight markers are shown in Appendix A.

**Figure 3 cancers-15-01843-f003:**
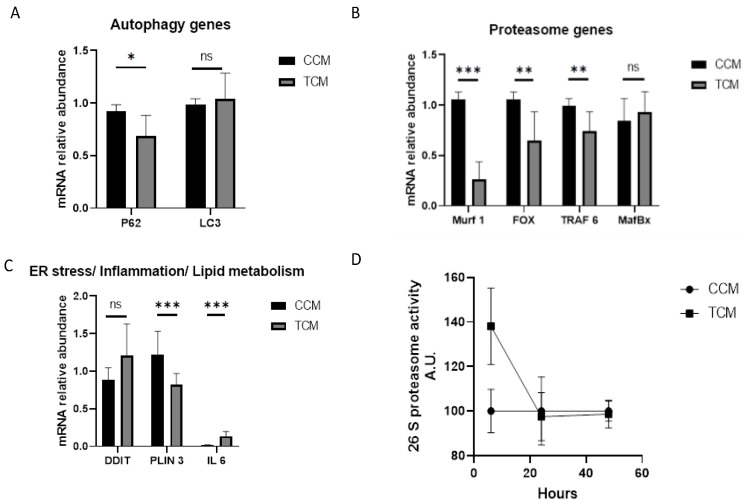
Change in mRNA expression in human primary myotubes incubated with either control-conditioned media (CCM) (n = 3 replicates) or with tumor-conditioned media (TCM) (n = 3 replicates). (**A**–**C**) Change in mRNA expression in human differentiated myotubes incubated for 48 h with either CCM or TCM. Gene expression in each condition was normalized to GAPDH expression. *: *p* < 0.05, **: *p* < 0.01, and ***: *p* < 0.001. (**D**): Time–course of proteasome activity in human-differentiated myotubes incubated with either CCM or TCM for different time-points (6 h, 24 h, and 48 h). ns: non significant.

**Figure 4 cancers-15-01843-f004:**
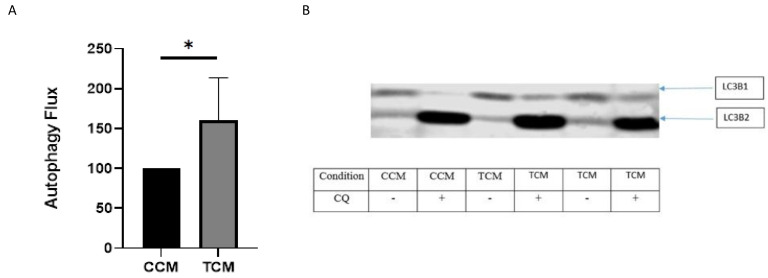
(**A**) Autophagy in human primary myotubes incubated with either control-conditioned media (CCM) or with tumor-conditioned media (TCM). TCM: Tumor-conditioned media made from UT-SCC-5 and UT-SCC-60A cancer cell lines. CCM: control-conditioned media. CQ: chloroquine. For experiments with conditioned media, n = 3 replicates. *: *p* < 0.05. Level of the autophagy flux was defined at a level of 100 arbitrary unit in one replicate to enable comparison between experiments. The uncropped blots and molecular weight markers are shown in Appendix A. (**B**) Illustration of a western blot used for the quantification of the autophagy flux shown in Figure 4A.

**Figure 5 cancers-15-01843-f005:**
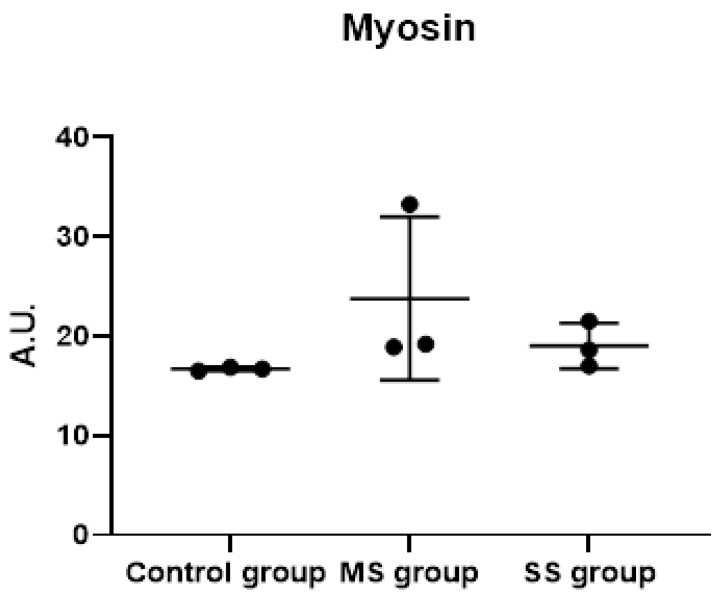
Myosin expression (heavy chain) in human primary myotubes incubated with either a mix of sera from cancer patients with mild or severe sarcopenia or from control participants. Mean ± SD of myosin expression (heavy chain) in myotubes incubated with either a mix of sera from cancer patients with severe sarcopenia (SS group), a mix of sera from cancer patients with mild sarcopenia (MS group), or a mix of sera from the control group. There were no significant differences found. A.U.: absorbance unit. For each condition, n = 2 replicates. The uncropped blots and molecular weight markers are shown in Appendix A.

**Figure 6 cancers-15-01843-f006:**
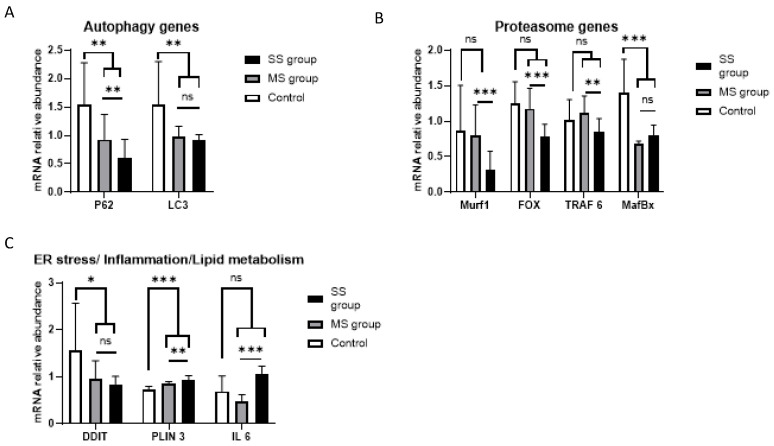
mRNA expression of human primary myotubes incubated for 48 h with a mix of sera from either control patients (control), HNC patients with severe sarcopenia (SS group), or HNC patients with mild sarcopenia (MS). For each condition, n = 2 replicates. Fold change in mRNA expression of human-differentiated myotubes incubated for 48 h with a mix of sera from either control patients (control), HNC patients with severe sarcopenia (SS group) or HNC patients with mild sarcopenia (MS). Gene expression in each condition was normalized to GAPDH expression. *: *p* < 0.05, **: *p* < 0.01, and ***: *p* < 0.001. (**A**) Genes related to the autophagy process. (**B**) Genes related to proteasomal activity. (**C**) Genes related to lipid metabolism, endoplasmic reticulum, or inflammatory processes.

**Figure 7 cancers-15-01843-f007:**
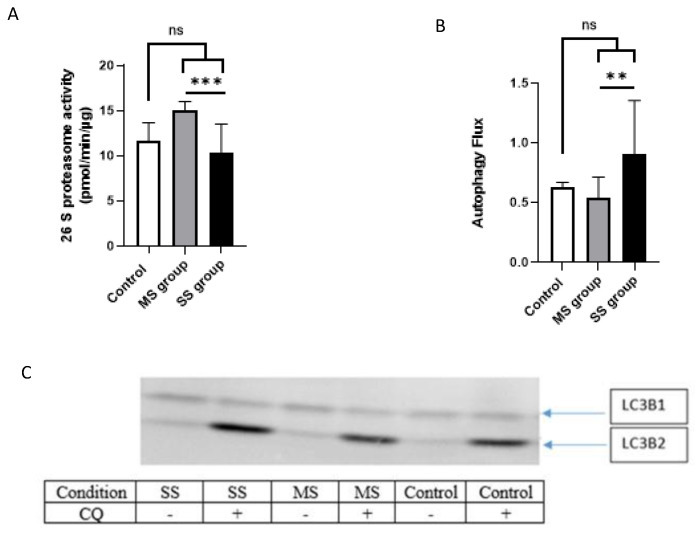
Autophagy and proteasome activity in human primary myotubes treated with HNC-patient tumor-conditioned media. (**A**): Proteasome activity human-differentiated myotubes incubated for 48 h with a mix of sera from either control patients (control), HNC patients with severe sarcopenia (SS group), or HNC patients with mild sarcopenia (MS). (**B**): Autophagy flux in differentiated myotubes incubated with mix of sera from control or cancer patients (MS and SS groups). The uncropped blots and molecular weight markers are shown in Appendix A. (**C**) Illustration of a blot used in this study for the experiment shown in B. For experiments made with mixed sera from HNC patients, n = 2 replicates. ** *p* < 0.01, ***: *p* < 0.001.

**Table 1 cancers-15-01843-t001:** Characteristics of the patients included in the study.

	Cancer (Group 1)	Control (Group 2)	*p*-Value
	Cancer (n = 20)	Control (n = 14)	*p*
Men (%)	19 (95)	1(4)	<0.001
Age (Years)	60 ± 8	46 ± 14	0.003
Weight (kg)	66.1 ± 11.4	82.7 ± 21.4	0.007
BMI (kg/m^2^)	22.0 ± 3.4	30.4 ± 7.0	0.003
3-month weight loss (%)	5.6 ± 5.7	−0.6 ± 1.4	<0.001
Handgrip strength (kg)	33.6 ± 10.9	28.7 ± 7.7	0.23
Kyle fat-free mass index (kg/m^2^)	18.3 ± 2.4	21.0 ± 7.1	0.15
Janssen muscular mass index (kg/m^2^)	10.0 ± 1.6	9.4 ± 1.2	0.18
L3MMI (cm^2^/m^2^)	43.5 ± 7.8	NA	NA
Serum albumin level (g/L)	35 ± 5.9	38.2 ± 3.8	0.14
Serum prealbumin (mg/mL)	0.22 ± 0.08	0.24 ± 0.04	0.34
Nutritional risk index	92.5 ± 9.8	100.0 ± 5.7	0.02
SPPB	9.5 ± 3.6	11.6 ± 0.84	0.05

**Table 2 cancers-15-01843-t002:** Tumor site and stage of disease in the cancer group. CUP: carcinoma of unknown primary.

		n (%)
Site	Hypopharyngeal Oral Oropharyngeal Laryngeal CUP	7 (35%) 4 (20%) 5 (25%) 3 (15%) 1 (5%)
Disease stage	I II III IVa IVb IVc	0 2 (10%) 1 (5%) 11 (45%) 3 (15%) 3 (15%)

**Table 3 cancers-15-01843-t003:** Clinical characteristics of the control group.

Non-malignant neck tumor	5 (36%)
Thyroid disease (nodules or thyroiditis)	9 (64%)
Surgery management	
Thyroidectomy	9 (64%)
Parotidectomy	3 (21%)
Paraganglioma	1 (7%)
Submandibular gland removal	1 (7%)

**Table 4 cancers-15-01843-t004:** Comparison of the effect of tumor-conditioned media and patient’s serum on human myotubes.

Biological Marker	Tumor Conditioned Media vs. Control Conditioned Media	Serum Patient Conditioned Media (Severely Sarcopenic Patient vs. Mild Sarcopenic Patient)
Myotubes size	↘	=
P62	↘	↘
LC3	=	=
Murf 1	↘	↘
Fox	↘	↘
TRAF 6	↘	↘
MafBx	=	=
DDIT	=	=
PLIN 3	↘	↗
IL 6	↗	↗
Proteasome activity	=	↘
Autophagy/lysosome pathway activity	↗	↗

↗: Significant increase in the biological marker; ↘: Significant decrease in the biological marker; =: no significant difference.

## Data Availability

All data could be obtained on demand from the first author (N.S.).

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
