# Peer review of "Conditioned Media from Head and Neck Cancer Cell Lines and Serum Samples from Head and Neck Cancer Patients Drive Catabolic Pathways in Cultured Muscle Cells"

_cancers, 2023, doi:10.3390/cancers15061843_

Round 1

Reviewer 1 Report

The authors have written a paper on how substances secreted from tumor cell lines from neck and from patience with head and neck cancer could influence skeletal muscle loss in vitro. It is a strength for the work that in vivo samples are included. The paper is overall well written, and the results somewhat support the conclusion “that the atrophy observed in HNC patients cannot be solely explained by a deficit in food intake”. However, there are a few concerns addressed below:

Major:

·      The relevance of the C2C12-cells is unsure, since there are similar effects on the human myotubes. C2C12 are mice cells and have a higher metabolism and growth rate than humans myotubes that may influence the results. This should be addressed in the manuscript.

·      It is unclear how the control media for the cell experiments is made. It seems like it is made based on the described media, and in that case is a weakness of the study. A correct control would be to expose the media to a healthy, corresponding cell line from the neck to be sure that the results is not a generic effect. This should be clarified and discussed, also since the conditioned medias from the human studies have a corresponding control group.

·      There is a significant difference in sex, average weight and BMI (22 vs 30) between the control group and the patient cohort.  This is a weakness of this study and should be addressed. Especially, possible implications of different hormone and metabolic profile on muscle loss.

·      A discussion on the difference between media from in vivo and from cell cultures is lacking. Especially, the possible speed of degradation of target substances in vivo vs in vitro would be relevant.

·      Figure 5 and line 440-441: to evaluate whether protein synthesis rate is changed and to claim that there are metabolic differences, the authors should study the incorporation and degradation of a protein in the myotubes. This can be done with radiolabelled trace studies. Other genes more specific for protein synthesis should also be analysed (e.i. MYH2, mTOR, MYOD, MAP8, NFkB), especially MYH2 would be relevant since it previously have been shown to be regulated in skeletal muscle after exposure to conditioned media from cancer cells.

Minor:

·      Abstract – HCN should be explained at first occurrence

·      TCM and CCM – should be deduced in the beginning of the result section to increase the reader friendliness

·      Figure 1 and 2: it is hard to understand how myosin is measured from the text (line 365), figure 1and the legend. This needs to clarified. 

·      Lacking number of experiences included (n) on figure 3, 6, 7 and 9? This should be corrected.

·      Typo: line 57, line 266

Author Response

The authors have written a paper on how substances secreted from tumor cell lines from neck and from patience with head and neck cancer could influence skeletal muscle loss in vitro. It is a strength for the work that in vivo samples are included. The paper is overall well written, and the results somewhat support the conclusion “that the atrophy observed in HNC patients cannot be solely explained by a deficit in food intake”. However, there are a few concerns addressed below:

Major:

  • The relevance of the C2C12-cells is unsure, since there are similar effects on the human myotubes. C2C12 are mice cells and have a higher metabolism and growth rate than humans myotubes that may influence the results. This should be addressed in the manuscript.

Response: This cell line has only been used in the first experiment presented in our article. Thereafter we have only used human primary cells. To remove confusion, we will use the term “human primary myotubes” in all our figure legends.

As the relevance of using c2c12 is a valid question, we have added in the text the following sentence: “C2C12 cell lines is a commonly used model to study muscle cell metabolism in vivo despite the fact that this is a non-human cell line[62] »

  • It is unclear how the control media for the cell experiments is made. It seems like it is made based on the described media, and in that case is a weakness of the study. A correct control would be to expose the media to a healthy, corresponding cell line from the neck to be sure that the results is not a generic effect. This should be clarified and discussed, also since the conditioned medias from the human studies have a corresponding control group.

We totally agree with the comment and this is a weakness of the study. we add line 247: “without cell” and we discuss this in a limitation paragraph that we add at the end of the discussion.

  • There is a significant difference in sex, average weight and BMI (22 vs 30) between the control group and the patient cohort.  This is a weakness of this study and should be addressed. Especially, possible implications of different hormone and metabolic profile on muscle loss.

We totally agree with the comment and this is another weakness of the study. This is discussed in a limitation paragraph of the discussion. However, the most interesting observations we have made were between the mild sarcopenic and severely sarcopenic, which remove the sex bias as these 2 population of patients were mostly male.

  • A discussion on the difference between media from in vivo and from cell cultures is lacking. Especially, the possible speed of degradation of target substances in vivo vs in vitro would be relevant.

We agree with this comment and comment on it in our limitation paragraph.

  • Figure 5 and line 440-441: to evaluate whether protein synthesis rate is changed and to claim that there are metabolic differences, the authors should study the incorporation and degradation of a protein in the myotubes. This can be done with radiolabelled trace studies. Other genes more specific for protein synthesis should also be analysed (e.i. MYH2, mTOR, MYOD, MAP8, NFkB), especially MYH2 would be relevant since it previously have been shown to be regulated in skeletal muscle after exposure to conditioned media from cancer cells.

We agree with the reviewer, our result on phosphorylated Akt led to an overinterpretation on our result on anabolism. Extra protein like p70s6k and their phosphorylation level would be needed to conclude. Concerning the MYH2 that the reviewer points out, I could only find paper related to the differentiation process. Which we haven’t studied here as our myotubes were all differentiated in the same media before incubation with experimental treatments.

If the reviewer agrees we could remove these results on akt from our paper and only present it as unshown data in the discussion.

Limitation paragraph :

Our study has some limitations that need to be addressed. In our conditioned media experiments, we have used a no cell control approach, a better control would have been to expose the myotubes to a conditioned media made after incubation for 24 h hour with a non-tumorigenic cell line corresponding to squamous cell carcinoma. With our approach, it is difficult to confirm that the effect of the TCM was only due to tumor secretions and not by the depletion of the media component by tumor cells during the 24 h incubation. However, the TCM was made from the tumor CM diluted with fresh media and supplemented with 2% serum as we used when we differentiated the myotubes with a media change every 48h. A second limitation of this study is that our control group does not match with the patient group in term of age, sex and body composition. Our results comparing control and cancer group could therefore be an effect of the difference in hormone and metabolic profile between the two groups. Due to that reason, we have chosen to look within our cancer group at the impact of patients´ muscle mass status on the in vitro effect of their serum on human myotubes. Furthermore, we saw that our most intriguing results were between the mild sarcopenic and severely sarcopenic HNC patients, where no difference in age, sex and body composition could explain the differences observed. The third limitation of our study is the comparison between experiments using tumor conditioned media and patients´ serum.  For the patient we used a concentration of 10% serum, a concentration much higher than the concentration of fetal bovine serum (2%) use during the tumor CM experiments. This might be a confounding factor explaining the discrepancy observed between the impact of TCM on myotubes atrophy that was not observed with patients’ serum.

Minor:

  • Abstract – HCN should be explained at first occurrence.
  • TCM and CCM – should be deduced in the beginning of the result section to increase the reader friendliness

We thank the reviewer for this comment. We have made these changes to make the reading more easy and clear.

  • Figure 1 and 2: it is hard to understand how myosin is measured from the text (line 365), figure 1and the legend. This needs to clarified.  We have clarify the sentence by

“We checked whether the change in myotube average diameter (assessed by microscopy) corresponded to a change in myotube myosin content (assessed by western blot analysis).”

  • Lacking number of experiences included (n) on figure 3, 6, 7 and 9? This should be corrected. The number of independent replicate have been added
  • Typo: line 57, line 266. We have corrected these typo

Reviewer 2 Report

This is a study about conditioned media from head and neck cancer cell lines and serum samples from head and neck cancer patients that drive anabolic and catabolic pathways in cultured muscle cells. Two cell lines for in vitro ecperiments were used.

The paper is well written. However, some issues remain.

Female sex and younger age are predominant in the control group. This may represent an important bias for the study.

Please briefly describe how genes were selected for expression analysis.

26S proteasome activity was different at 0 hours (CCM versus TCM). Please discuss this and look for possible biases.

A table that summarizes and compares results with conditioned media and serum samples may help the readers.

Author Response

This is a study about conditioned media from head and neck cancer cell lines and serum samples from head and neck cancer patients that drive anabolic and catabolic pathways in cultured muscle cells. Two cell lines for in vitro ecperiments were used.

The paper is well written. However, some issues remain.

Female sex and younger age are predominant in the control group. This may represent an important bias for the study.

We agree with the reviewer about this important bias. We have addressed that issue in our limitation paragraph.

Please briefly describe how genes were selected for expression analysis. We have chosen to focus on the autophagy pathway by looking at the protein expression of LC3 and p62 and at their gene expression as p62 can be highly regulated at the transcriptomic level especially during severe starvation. The proteasome pathway was monitored by looking at expression levels of the E3 ligases, MURF1 and atrogin1, that have been shown to be increased in many modell of cachexia and muscle atrophies. We also looked at TRAF6 and FOXO3 expression as they are transcription factors that can regulate autophagy and proteasome and that have been shown to be dysregulated in certain model of cachexia.

For the DDIT, also known as CHOP, it was to monitor the ER stress, that could have been a confounding factor. The IL-6 was used as a marker of inflammation, as an increase of the gene in muscle has been commonly find in various model of cachexia and muscle atrophy. The plin3 gene was to look at lipolysis and we choose to keep this marker as it revealed interesting observations.

26S proteasome activity was different at 0 hours (CCM versus TCM). Please discuss this and look for possible biases.

It is a misunderstanding, this activity was not measure at baseline but after 6h incubation.

A table that summarizes and compares results with conditioned media and serum samples may help the readers.

We explain in the limitation paragraph that this is not the aim of the study and that this comparison could be hazardous because of the difference in serum concentration between these two experimental conditions.

Round 2

Reviewer 1 Report

The revision addresses my previous concerns so I have no further comments.

Author Response

Thanks for your comments you can find in an attached file our respons.

Best

Reviewer 2 Report

Please describe how genes were selected for expression analysis also in the manuscript. 

A table that summarizes and compares results with conditioned media and serum samples should be added.

Author Response

(The authors gave the same response as above.)

Round 3

Reviewer 2 Report

Thanks for improving the manuscript.